# Concentration, Spatial Distribution, Contamination Degree and Human Health Risk Assessment of Heavy Metals in Urban Soils across China between 2003 and 2019—A Systematic Review

**DOI:** 10.3390/ijerph17093099

**Published:** 2020-04-29

**Authors:** Shuangmei Tong, Hairong Li, Li Wang, Muyesaier Tudi, Linsheng Yang

**Affiliations:** 1Key Laboratory of Land Surface Pattern and Simulation, Institute of Geographical Sciences and Natural Resources Research, Chinese Academy of Sciences, 11 A Datun Road, Beijing 100101, China; tongsm.19b@igsnrr.ac.cn (S.M.T.); wangli@igsnrr.ac.cn (L.W.); muys.14b@igsnrr.ac.cn (M.T.); yangls@igsnrr.ac.cn (L.Y.); 2College of Resources and Environment, University of Chinese Academy of Sciences, Beijing 100049, China; 3College of Tourism and Historical Culture, Liupanshui Normal University, Liupanshui 553004, China

**Keywords:** urban soil, heavy metals, spatial distribution, contamination degree, health risk

## Abstract

This study provides an overview of the studies of heavy metal pollution regarding As, Cd, Cr, Hg, Pb, Cu, Zn and Ni in the urban soils throughout 71 cities of China, based on data from online literature, during the period 2003–2019. The concentrations, spatial distributions, contamination degrees and health risks of heavy metals in the urban soils were evaluated. The results demonstrated that the mean values of eight heavy metals all exceeded the soil background values in China, and the kriging interpolation method showed that the hot-spot cities with heavy metal contamination in urban soils were mainly concentrated in the southwest, southcentral, southeast coast, northcentral and northwest regions of China. The geoaccumulation index (*Igeo*) indicated that Hg and Cd were at moderate contamination levels and that the levels of the other six metals did not appear contamination. The pollution index (*PI*) showed that Cd and Hg reached high contamination levels, and the other metals reached moderate contamination levels. The integrated pollution index (*IPI*) and potential ecological risk index (*PRI*) indicated that the integral urban soils in the study areas ranked high contamination levels and moderate ecological risk degree, respectively, and Cd and Hg should be labeled as priority metals for control in the urban soils around China. The human health risk assessments for the heavy metals indicated that ingestion was the dominant exposure pathway for having adverse effects on human health. The mean Hazard index (*HI*) values of eight heavy metals all showed that adverse effects on human health were unlikely, and the mean carcinogenic (*CR*) values of As, Cr and Ni for children and adults all suggested an acceptable carcinogenic risk to human beings. In addition, children exposed to these heavy metals faced more serious non-carcinogenic and carcinogenic health threats compared to adults. The results could provide valuable information for demanding the better control of heavy metal pollution and mitigation of the adverse effects on residents by environmental regulators in national urban regions.

## 1. Introduction

As a crucial component of urban ecosystems, soil plays a principal role in biochemical transformation, the cycling of elements, supporting plants and many recreational activities. In addition, it is meanwhile considered to be a mixture of heavy metals, mineral constituents, organic matter (humus), living organisms, air and water [1,2,3]. Urban areas are the hot-spots for environmental hazards at multiple scales, as a result of increased population, industrial growth and vehicular transport increase [4]. With the rapid development of urbanization and the increase in the urban population, the disturbance by human activities of the urban soil environment has led to varying degrees of deterioration in urban soil environment quality. Urban soils, acting as a reservoir of contaminants, are excellent indicators of pollution [5], and heavy metals in soil are considered as important indicators to monitor the impact of human activities on soil environmental quality [6,7,8].

Owing to the toxic effects, long-term persistence and bio-magnification traits, even at low concentrations, heavy metal pollution has attracted widespread attention and heavy metals are considered to be the most important pollutants among the multitudinous soil contaminants [9,10,11]. Urban soils can serve as recipients of large amounts of heavy metals from multiple sources. Heavy metals that accumulate in soils simultaneously originate from natural and other anthropogenic sources [6,12]. Urban soils differ greatly from natural ones, as they are more strongly influenced by anthropogenic activities [1,13]; thus, anthropogenic sources are considered to be the primary source cause of soil contamination; for instance, industrial waste, automobile exhausts and domestic waste are seen as major causes of the increased content of potentially toxic elements (PTEs), such as Pb, Cd, Cu and Zn [4,14]. As a result, they are more prone to containing and accumulating high concentrations of heavy metals in comparison with the natural soils [5]. The accumulation of heavy metals in soils can inevitably affect environmental quality, such as through urban soil, water and crop contamination. Pollutants can be transferred into the human body through the food chain, ultimately posing direct or indirect health hazards to the human beings in the long-term [15,16]. For example, heavy metals accumulated in the tissues and internal organs of human body can affect the central nervous system and may act as cofactors, initiators or promoters of other diseases [3,17]. Furthermore, exposure to mixed metals can result in numerous adverse health effects on humans due to synergistic interactions, even when concentrations of the individual metals are below their ecotoxicological benchmark levels [18]. The adverse effects of heavy metals on human health are mainly conferred through three pathways: ingestion, inhalation and dermal contact absorption; numerous studies have shown that ingestion is the primary exposure pathway for human health risks; in addition, children are especially susceptible to health risks from heavy metal toxicity [17,19,20,21].

As the largest developing country in the world, since 1978, China has witnessed a dramatic growth in urbanization along with unprecedented economic growth [22]; however, heavy metal pollution has become a serious environmental problem in different functional regions, such as urban soils, urban road dust and agricultural soils, with rapid industrialization and urbanization over the past two decades [23,24]. The national soil pollution survey bulletin of China in 2014 showed that the proportions of slight, mild, moderate and severe pollution spots around China were 11.2%, 2.3%, 1.5% and 1.1%, respectively. Therefore, to provide guidance for the prevention and remediation of soil pollution in China over the next 30 years, the Chinese government issued a regulation in 2016, the “Soil Contamination Prevention and Control Action Plan” [9]. 

Studies of urban soils in China started in the 1980s, whereas discussions of the impact of urbanization on soil resources in China have been more in recent years; in addition, more than 100 cities have been studied [22,25,26]. In China, numerous research studies have been conducted on metal concentrations, spatial distributions, contamination assessments, source identification and health risk assessments, while the vast majority of studies have focused on heavy metals in the urban soil of a single city [19,27,28,29,30]. In recent decades, more research has paid attention to the extensive range of heavy metal pollution in urban soil across China. For example, Pan et al. reviewed heavy metal pollution levels and performed a health risk assessment of urban soils in 32 Chinese cities [31]. Wei and Yang reviewed heavy metal contamination in urban soils, urban road dusts and agricultural soils from China [32]. Luo et al. examined trace metal concentrations, pollution levels and sources identification in 21 Chinese cities [1]. Zhang et al. assessed the spatial distribution of metal pollution in the soils of Chinese provincial capital cities [33]. Guo et al. reported the spatial distribution and pollution assessment of heavy metals in urban soils from southwest China [6]. However, there was very limited report in the aforementioned studies regarding the assessment of the urban soils throughout China in vast quantity and scope; thus, carrying out a further, comprehensive and national scale study across China is urgently needed.

Therefore, this study focuses on the concentrations, contamination, spatial distribution and human health risks of heavy metals in the urban soils throughout China on a national scale, based on the online literature data. The main objectives of this study are (a) to determine the concentrations and spatial distribution of As, Cd, Cr, Hg, Pb, Cu, Zn and Ni—eight heavy metals—in urban soils; (b) to evaluate the pollution characteristics of the heavy metals using the geoaccumulation index (*Igeo*), pollution index (*PI*) and integrated pollution index (*IPI*); (c) to discriminate the possible hot-spots of heavy metal contamination in urban soils; (d) to investigate the potential ecological risk degree and detect the risk factors that contribute most to national urban soil contamination, using the potential ecological risk index (*PRI*) ; and (e) to evaluate the human health risks to the child and adult communities through different exposure pathways. 

## 2. Data Source and Research Methods 

### 2.1. Data Source

#### 2.1.1. Search Method

The study chose As, Cd, Cr, Hg, Pb, Cu, Zn and Ni—eight elements—as the target contaminants, all of which were listed as priority pollutants for control by the United States Environmental Protection Agency (USEPA). The information sources for the systematic literature review were located in three databases: China National Knowledge Internet (CNKI), Web of Science and Google Scholar. We used three categories of keyword by connecting the same group with “OR” and combining the different group with “AND”, which comprised the following: (a) metals—heavy metals, metal element, trace element, and metallic element; (b) status—concentration, levels, contents, contamination, pollution, degree, quality, spatial distribution, health risk, assessment, evaluation, and condition; (c) settings—urban, soil, city, capital, metropolitan, district, region, functional zone, area, China, and Chinese. The eligibility criteria comprised the following: (a) the study area was restricted to the urban areas of China, and the research medium was soil; (b) the papers were published between 2003 and 2019 and in the most recent years; (c) the published papers included as many of the eight elements of the review as possible, with at least three elements involved; (d) the papers focused on urban soils with surface layers with a depth of 0–20 cm; (e) the studies could well reflect the overall soil pollution characteristics of the urban environment, including no less than two functional regions (such as an industrial region, a residential district, etc.); (d) the studies were conducted with scientific sampling, analytical methods and strict quality assurance or quality control procedures; and (f) the papers were published in full text, excluding abstracts or news reports.

#### 2.1.2. Screening Method

We tried to select the studies published during the most recent years, and those can best reflect the overall soil pollution characteristics of the urban situation, from the three databases of China National Knowledge Internet (CNKI), Web of Science and Google Scholar according to the search strategy from the period 2003–2019. Furthermore, we tried to obtain as many data as possible, to cover every province and present the overall contamination condition of urban soils throughout the country. The soil samples in the selected articles were digested with mixed acids such as HF + HCLO_4_ + HNO_3_, HNO_3_ + H_2_O_2_ or HNO_3_ + HF + HCL. Afterwards, the total concentrations of As, Cd, Cr, Hg, Cu, Pb, Zn and Ni were determined by ICP, ICP-MS, ICP-OES, ICP-AES or AAS. All of the sample processing and analytical methods were controlled with strict quality assurance and are accepted by the scientific community. In addition, abnormal values were eliminated. The non- qualified articles were excluded by means of title review, abstract review and full text review based on the eligibility criteria. Finally, a total of urban soil heavy metal pollution datasets from 71 Chinese cities, including 10,071 sample sites covering almost every province across China—with the exception of Taiwan—were collected. The distribution of the cities is presented in Figure 1. 

### 2.2. Research Methods

#### 2.2.1. Geoaccumulation Index (*Igeo*)

Since 1969, the geoaccumulation index (*Igeo*) has been commonly used as a geochemical criterion to evaluate the contamination degree of a single element in environmental sediments or soils. The equation to calculate *Igeo* values is shown as follows [34]: *Igeo* = *log*_2_ (*C_n_*/1.5*B_n_*)(1)
where *C_n_* represents the concentration of the measured element in the sediment and *B_n_* is the geochemical background value of the given metal. The constant 1.5 is a background matrix correction factor considering the natural fluctuations influenced by lithogenic effects. In this study, the geometric means of background values of the corresponding metal in the control district in which the city is located were chosen as the background values according to the background values of the elements in the soils of China [35]. The *Igeo* is divided into seven levels: *Igeo* ≤0, uncontaminated; 0 < *Igeo* ≤ 1, uncontaminated to moderately contaminated; 1 < *Igeo* ≤ 2, moderately contaminated; 2 < *Igeo* ≤ 3, moderately to heavily contaminated; 3 < *Igeo* ≤ 4, heavily contaminated; 4 < *Igeo* ≤ 5, heavily contaminated to extremely contaminated; and 5 ≤ *Igeo*, extremely contaminated.

#### 2.2.2. Pollution Index

The pollution index (*PI*) is defined as the ratio of the heavy metal concentration to the geometric mean of background concentrations, which is generally used to calculate the pollution level of individual elements; the integrated pollution index (*IPI*) is defined as the mean value of all of the *PI* of all considered metals. In addition, the *IPI* is used for the determination of heavy metal contamination in individual samples in the area, rather than revealing the general contamination degree of the whole area [11,36,37]. The calculation of *PI* is as follows:*PI* = *C_i_*/*B_i_*(2)
where *C_i_* represents the concentration of the measured metal *i* in the sediment, and *B_i_* represents the background value of corresponding metal *i* [35]. The *PI* value of each metal and the *IPI* value of each sample site are calculated and classified respectively as low contamination (*PI* ≤ 1.0), moderate contamination (1.0 < *PI*≤ 3.0) or high contamination (*PI* > 3.0); and low contamination (*IPI* ≤ 1.0), moderate contamination (1.0 < *IPI* ≤ 2.0), high contamination (2.0 < *IPI* ≤ 5.0) or extremely high contamination (*IPI* > 5) [11,32,36]. 

#### 2.2.3. Potential ecological Risk and Health Risk Assessment

The potential ecological risk index (*PRI*) was initially introduced by Hakanson, with the purpose of quantifying the ecological risk of one or multiple target contaminants in a specific environment medium [11,37,38], which reflected the side effects of toxic elements on the environment. The formulas for *PRI* are evaluated as follows: (3)$$E_{r}^{i}=T_{r}^{i}\times C_{f}^{i},C_{f}^{i}=C_{s}^{i}/C_{n}^{i}$$
(4)$$PRI=\sum_{i=1}^{n}E_{r}^{i}$$
where $C_{f}^{i}$ refers to the contamination coefficient of heavy metal *i*, $T_{r}^{i}$ is the biological toxic response factor for heavy metal *i*, $C_{s}^{i}$ is the measured concentration of heavy metal *i*, $C_{n}^{i}$ is the background value of heavy metal *i* [35], $E_{r}^{i}$ refers to the potential ecological risk factor of a single metal, and *PRI* refers to the potential ecological risk index of multiple elements. The toxic response factors $T_{r}^{i}$ for As, Cd, Cr, Hg, Pb, Cu, Zn and Ni are 10, 30, 2, 40, 5, 5, 1 and 5 respectively [39]. The classifications for ecological risk degree are presented as low ecological risk ($E_{r}^{i}$ < 40 or PRI < 150), moderate ecological risk (40 ≤ $E_{r}^{i}$ < 80 or 150 ≤ PRI < 300), considerable ecological risk (80 ≤ $E_{r}^{i}$ < 160 or 300 ≤ PRI < 600), high ecological risk (160 ≤ $E_{r}^{i}$ < 320 or 600 ≤ PRI) or very high ecological risk (320 ≤ $E_{r}^{i}$) [11,40].

Human health risk assessment is widely used to quantify the exposure of humans to chemical elements and carcinogenic and noncarcinogenic risks, with ingestion, inhalation and dermal contact as the three exposure pathways [15,36]. In this study, the exposure model is based on the method developed by the Environmental Protection Agency of the United States [41]. The calculation formulas for the average daily doses (*ADD*) (mg kg^−1^ d^−1^) of potentially toxic metals in adults and children via the three exposure pathways are as follows [42,43]:*ADD_ing_ = (C × IngR × CF × EF × ED)/(BW × AT)*(5)
*ADD_inh_ = (C × InhR × EF × ED)/(PEF × BW × AT)*(6)
*ADD_derm_ = (C × SA × CF × SL × ABS × EF × ED)/(BW × AT)*(7)
where *ADD_ing_*, *ADD_inh_* and *ADD_derm_* are the average daily doses of exposure to toxic metals through ingestion (mg kg^−1^ d^−1^), inhalation (mg kg^−1^ d^−1^) and dermal contact (mg kg^−1^ d^−1^) by the three pathways, respectively. The exposure parameters for the three models are listed in Table 1 and are based on the USEPA (2001) [44] and environmental site assessment guidelines in China (2009) [45]. 

The Hazard index (*HI*) was introduced to assess the overall potential noncarcinogenic risks induced by the toxic metals, the carcinogenic risk (*CR*) was regarded as the probability of an individual developing any type of cancer over the whole lifetime due to exposure to carcinogenic hazards [46,47]. In this study, eight heavy metals—As, Cd, Cr, Hg, Pb, Cu, Zn and Ni—were all considered to be existing noncarcinogenic risks; furthermore, the carcinogenic risks of As, Cr and Ni were evaluated due to the unavailable carcinogenic slope factors of other toxic metals. The calculations of the Hazard index (*HI*) and carcinogenic risks (*CR*) are as follows:(8)$$HI=\sum HQ_{i}=\sum ADD_{i}/RfD_{i}$$
(9)$$CR=\sum ADD_{i}\times SF_{i}$$
where *RfD* (mg kg^−1^ d^−1^) is the reference dose for each heavy metal, *SF* (mg kg^−1^ d^−1^) ^−1^ is the carcinogenic risk probability, and the parameter values of *RfD* and *SF* for the toxic metals are derived from the research conducted in China (Table 2) [15,48]. *HQ* refers to the hazard quotient generated for each element and exposure pathway and *HI* is the comprehensive noncarcinogenic risk equal to the total of *HQ*. When the value of *HI* is less than 1, adverse health effects are unlikely; when the value of *HI* is more than 1, adverse health effects may occur. If *CR* <10^−6^, the carcinogenic risk is considered to be negligible; if *CR* >10^−4^, there is high risk of developing cancer in human beings; and when 10^−6^ < *CR* < 10^−4^, there is an acceptable risk to human beings [17,46,49].

#### 2.2.4. Statistical Analysis 

The mean values, minimum values, maximum values, standard deviations, coefficients of variation (CV), *Igeo, PI, IPI*, $E_{r}^{i}$, *PRI, HI* and *CR* of heavy metals were calculated and summarized in Microsoft Excel (Microsoft Inc., Redmond, Washington, USA). The spatial distributions of heavy metals were determined using the kriging interpolation method, performed by the geostatistical analysis GIS software ArcGIS 10.1 (ESRI Inc, Redlands, California, USA). Box-plots for the *Igeo*, *PI, IPI*,$E_{r}^{i}$ and *PRI* of heavy metals were created by the software package SPSS 25.0, and the distributions of the *IPI*, *PRI* and *CR* of heavy metals were created by the ArcGIS 10.1 software.

## 3. Results and Discussion

### 3.1. Heavy Metal Concentrations and Spatial Distributions 

The minimum, maximum, mean, median, standard deviation and coefficient of variation statistical characteristics and the spatial distributions of the eight heavy metals in the urban soils of 71 cities, at the national scale in China, are presented in Table 3 and Figure 2, respectively. As shown in Table 3, the concentrations of the heavy metals were in the following ranges: As: 3.82–32.80 mg/kg, with a mean value of 11.53 mg/kg; Cd: 0.10–6.90 mg/kg, with a mean value of 0.79 mg/kg; Cr: 15.32–378.86 mg/kg, with a mean value of 77.86 mg/kg; Hg: 0.04–0.77 mg/kg, with a mean value of 0.27 mg/kg; Pb: 7.52–409.20 mg/kg, with a mean value of 60.26 mg/kg; Cu: 13.55–430.00 mg/kg, with a mean value of 47.72 mg/kg; Zn: 26.00–374.47 mg/kg, with a mean value of 128.21 mg/kg; and Ni: 8.41–361.00 mg/kg, with a mean value of 37.99 mg/kg. All of the mean values exceeded the soil background values of China [35]; in particular, Cd and Hg were 8.14 and 4.15times higher than the national background, respectively, indicating the influence of urbanization and industrialization on urban soil pollution and that the pollutants’ influence on the soil environment is serious [26,28]. The results showed that the median concentrations of all the elements were slightly lower than their mean concentrations. In addition, the CV of heavy metals in urban soils decreased in the following order: Cd > Ni > Cu > Pb > Cr > Hg > Zn > As, all of which manifested a high degree of variability and a much stronger nonhomogeneous distribution of concentrations, due to anthropogenically emitted heavy metals, with increasing CV values, especially for Cu, Ni and Cd [6,15,47].

In order to identify the spatial distributions of the heavy metal concentrations in reviewed urban soils across China, the ordinary kriging interpolation method was used to obtain the spatial distribution patterns of the elements, performed by the GIS software (ArcGIS 10.1) (Figure 2). It should be mentioned that any interpolation methods involve uncertainty, and the results from specific sites should be taken as the expected possible values but not the true values [115]. As shown in Figure 2, the element of As presented high concentrations in the south region and low concentrations in the north, northwest and northeast regions; the hot−spot locations were concentrated in several cities, including Guiyang, Zhuzhou, Changsha and Lanzhou. The high concentrations of Cd were mainly distributed in the southcentral and northwest region, and the hot-spots were located in Kashi, Wuhan and Changsha cities. For Cr, the high concentrations were mainly concentrated in the northcentral, and north regions, including Panzhihua, Ganzhou, Xuzhou, Changsha, Jiaozuo, Yongkang, Xining and Yinchuan hot−spot cities. The high concentrations of Hg were mainly distributed in the northcentral and east coastal regions, with the hot-spots located in Shangluo, Guangzhou, Zhangzhou, Fuzhou and Suzhou cities. The element of Pb presented high concentrations in the southeast and northwest regions, with the hot-spot cities being located in Hong Kong, Dongguan, Guangzhou, Ganzhou, Baoji, Tongchuan and Shangluo. Similar distribution patterns manifested for Cu and Ni, both of which had hot-spots situated in Kunming, Panzhihua, Guiyang and Jinchang cities. As for Zn, the high concentrations were concentrated in Chengdu, Guiyang, Changsha, Baoji, Lanzhou, Luoyang, Shenyang, Changji, Shanghai and Urumqi cities. 

On the whole, the hot-spot cities for heavy metals in urban soils were mainly concentrated in the southwest, southcentral, southeast, east coast, northwest and northcentral regions of China, which should be labeled as priority regions for controlling heavy metal contamination. It was indicated that the southern part was more contaminated than the northern part, which may be associated with the high geochemical background in the southwest regions and long-term industrial activities. For example, Yunnan Gejiu Tin capital, Lanping Zinc capital, Sichuan Panzhihua V-Ti magnetite, Guizhou Liupanshui coal mine, Gansu Jinchang nickel capital, Hunan Shizhuyuan polymetallic mine, Guangxi Dachang Sb polymetallic mine, Guangdong Dabaoshan polymetallic mine, and Fujian Youxi Pb-Zn mine are all located in the highly-contaminated regions, which could generate other accompanying heavy metals such as Cd and Cr [116]. Furthermore, the increasing urban populations and manufacturing industries activities may also be significant contributors to the pollution [27,47,117]. In addition, since the data collected in Xizang, Xinjiang and Gansu provinces had less published literature and several abnormal values were eliminated in Qingdao, the authentic contamination levels of that region might not have been accurately determined. 

### 3.2. Assessment of Heavy Metals Pollution in Urban Soils Using Geochemical Indicators

The Box-plots of the Igeo, PI, IPI, $E_{r}^{i}$ and *PRI* of the heavy metals in urban soil around China are presented in Figure 3; as shown in Figure 3, the *Igeo*, *PI, IPI*,$E_{r}^{i}$
*PRI* values of the heavy metals varied significantly across different sites. The *Igeo* values for As, Cd, Cr, Hg, Pb, Cu, Zn and Ni ranged from −1.82 to 0.78, −1.63 to 5.19, −2.95 to 1.99, −0.43 to 3.69, −1.95 to 3.67, −1.29 to 3.57, −1.76 to 1.85 and −2.25 to 2.77, respectively, and the mean values decreased in the following order: Cd (1.35) > Hg (1.32) > Pb (0.26)> Cu (0.18) > Zn (0.16) > Cr (−0.51) > Ni (−0.56) > As (−0.59). The mean pollution levels of Hg and Cd represented moderate contamination levels, Pb, Cu and Zn indicated uncontaminated to moderately contaminated levels, and the levels of Cr, Ni and As represented non−contamination levels, based on the *Igeo* classification criterion. As was shown in Figure 4, As manifested at uncontaminated and uncontaminated to moderately contaminated levels in 82.98% and 17.02% of sites, respectively. Cd manifested the most serious contamination in the study area, manifesting at the different levels in the following proportions of sites: uncontaminated, 16.98%; uncontaminated to moderately contaminated, 28.30%; moderately contaminated, 28.30%; moderately to heavily contaminated, 11.32%; heavily contaminated, 11.32%; and extremely contaminated, 3.77%. The *Igeo* values of Cd in the cities of Changsha and Kashi reached 5.19 and 5.14, respectively, all of which denoted extreme contamination. For Cr and Zn, uncontaminated, uncontaminated to moderately contaminated and moderately contaminated levels represented 75.41%, 19.67% and 4.92%; and 43.94%, 40.91% and 15.15% of sites, respectively. Hg manifested at different levels of contamination according to the following proportions: uncontaminated, 6.45%; uncontaminated to moderately contaminated, 32.26%; moderately contaminated, 38.71%; moderately to heavily contaminated, 16.13%; and heavily contaminated, 6.45%. The higher *Igeo* values for Hg were located in Shangluo (3.69) and Baiyin (3.17) cities, all of which reached heavily contaminated levels. Pb manifested contamination levels with uncontaminated, uncontaminated to moderately contaminated, moderately contaminated, moderately to heavily contaminated and heavily contaminated levels in 45.07%, 36.62%, 14.08%, 1.41% and 2.82% of sites, respectively; and Baoji and Tongchuan cities showed heavily contaminated levels, the *Igeo* values of which reached 3.67 and 3.48, respectively. For Cu, contamination levels with uncontaminated, uncontaminated to moderately contaminated, moderately contaminated, and heavily contaminated levels were in 42.65%, 42.65%, 13.24% and 1.47% of sites, respectively; and the most serious contamination was located in Jinchang, which showed heavily contaminated levels, with *Igeo* values reaching 3.57. Ni had the most sites with uncontaminated levels, with uncontaminated, uncontaminated to moderately contaminated, moderately contaminated, moderately to heavily contaminated levels in 87.80%, 2.44%, 4.88% and 4.88% of sites, respectively.

The mean *PI* values decreased in the following order: Cd (7.07) > Hg (4.85) > Pb (2.45) > Cu (2.10) > Zn (1.91) > Ni (1.43) > Cr (1.28) > As (1.09) (Figure 3). The mean values of Cd and Hg reached high contamination levels, and the mean values of Pb, Cu, Zn, Ni, Cr and As reached moderate contamination levels, based on the *PI* classification criterion. The *PI* values of Cd in Changsha (54.76) and Kashi (52.83) manifested the highest contamination levels among all the cities; for Hg, the highest contamination levels were located in Shangluo, Baiyin and Shenyang cities, with *PI* values of 19.33, 13.50 and 10.54, respectively. For Pb, the highest contamination levels were located in Tongchuan and Baoji cities, with *PI* values of 16.70 and 19.12, respectively. The highest contamination levels of Cu and Ni were located in Jinchang city, with *PI* values of 17.84 and 10.26, respectively. For Zn and As, the highest contamination sites were located in Baoji, Jiaozuo and Shenyang cities, respectively, and the *PI* values of were 5.40, 5.94 and 2.58, respectively. All of the high *PI* value cities, for every metal mentioned in the former, reached high contamination levels, with the exception of the *PI* value of As in Shenyang, which reached moderate contamination levels.

The *IPI* values ranged between 0.83 and 10.69 among all of the cities, with a mean value of 2.73 (Figure 3), which showed that the mean contamination of all of the urban soils in the study areas was at a high contamination level, based on the *IPI* classification criterion. As shown in Figure 5, the low contamination levels were mainly distributed in the northern region, and the high contamination and extremely high contamination levels were mainly distributed in the northwest, north center, southwest, south center, east and east coast regions of China. The extremely high contamination levels of the cities were mainly located in Kashi, Changsha, Shangluo, Tongchuan, Luoyang, Jinchang, Baoji and Ganzhou, Lvliang, and Shenyang, the *IPI* values of which were 10.69, 9.99, 7.23, 7.16 7.10, 6.82, 6.38, 6.06, 5.56 and 5.47, respectively.

The mean $E_{r}^{i}$ values decreased in the following order: Cd (212.23) > Hg (194.17) > Pb (12.26) > As (10.91) > Cu (10.50) > Ni (7.13) > Cr (2.48) > Zn (1.91). Based on the $E_{r}^{i}$ classification criterion, the mean $E_{r}^{i}$ values of Cd and Hg showed high ecological risks degrees; the mean $E_{r}^{i}$ values of other six metals were below 40, representing low ecological risk degrees. As shown in Figure 4, the percentages of ecological risk degrees for As, Cr and Zn were at low ecological risk degree in 100% of sites; Pb, Cu and Ni manifested low ecological risk degrees in 97.18%, 98.53% and 95.12% of sites, respectively; for Cd, low, moderate, considerable, high and very high ecological risk degrees presented in 15.09%, 24.53%, 28.30%, 15.09% and 16.98% of sites, respectively; and Hg manifested moderate, considerable, high and very high ecological risk degrees in 25.81%, 25.81%, 35.48% and 12.90% of sites, respectively. Therefore, it could be inferred that Cd and Hg served as the predominant ecological risk factors amongst the eight metals, and contributed the most to the ecological risk degree of urban soils among all of the study cities.

The *PRI* values ranged between 17.40 and 1835.91 among all of the cities, with a mean value of 280.93 (Figure 3), which demonstrated that the mean ecological risk of all the urban soils in the study areas was moderate, based on the *PRI* classification criterion. As shown in Figure 5, the low ecological risk degrees and moderate ecological risk degrees were mainly distributed in the north of China, while the considerable ecological risk degrees and high ecological risk degrees were mainly distributed in the southwest, south center, north center, east and east coast regions of China, which presented similar distributions of the *IPI*. High ecological risk degrees were mainly located in Changsha (1835.91), Kashi (1631.60), Shangluo (994.86), Lvliang (671.64), Shenyang (809.63), Luoyang (737.56) and Baiyin (655.60).

On the whole, the *Igeo*, *PI* and $E_{r}^{i}$ values of Cd and Hg were all higher than those of the other six metals, and these manifested the highest contamination levels and ecological risk degrees among the eight metals; consequently, Cd and Hg should be labeled as the priority metals for control in the urban soils around China. Furthermore, according to the *IPI* and *PRI* values (Figure 5), the high contamination and ecological risk regions were mainly concentrated in the southwest, south center, north center, east and east coast regions of China, while the cities were mainly located in Kashi, Changsha, Shangluo, Tongchuan, Lvliang, Jinchang, Luoyang, Baoji, Ganzhou, Shenyang and Baiyin. 

### 3.3. Health Risk Assessment of Heavy Metal Contamination in Urban Soils

The assessment results of the mean values of non−carcinogenic and carcinogenic risks of each element, through the three exposure pathways, in urban soils are shown in Table 4. The mean HQ values of the eight metals for children and adults via ingestion (HQ_ing_), inhalation (HQ_inh_) and dermal contact (HQ_dermal_) all descend in the trend HQ_ing_ > HQ_dermal_ > HQ_inh_, showing that ingestion was the predominant exposure pathway to pose adverse effects on human health, which was consistent with previous research [15,19]. In addition, the HQ_ing_ and HQ_dermal_ values for children were higher than those for adults, while for the HQ_inh_ values, the opposite was true. The mean *HI* values of As, Cd, Cr, Hg, Pb, Cu, Zn and Ni for children and for adults were all less than 1, illustrating that adverse effects on human health were unlikely; in addition, they followed the decreasing trend for children As > Cr > Pb > Ni > Cd > Cu > Hg > Zn, and for adult Cr > As > Pb > Ni > Cd > Hg > Cu > Zn. The mean pollution levels of Hg and Cd represented moderately contaminated levels, whereas Cr and As were at uncontaminated levels, based on the *Igeo* classification criterion; however, Cr and As contributed most to the *HI* while Hg and Cd did less so, illustrating that the metals at uncontaminated levels should also be paid attention to. In term of the individual urban sites, the *HI* values of the eight metals for adults were all less than 1; however, some of the risk values for children exceeded 1—for example, the *HI* values of As in Changsha (1.45) and Nanning (1.02); Cr in Jiaozuo (2.29), Xining (1.10) and Ganzhou (1.04); and Pb in Baoji (1.60) and Tongchuan (1.39)—which demonstrated that heavy metals were unlikely to have adverse effects on adults, whereas adverse health effects may occur on children. Furthermore, the *HI* values for children were higher than those for adults, indicating that children are exposed more serious non−carcinogenic health threats compared to adults, which might be induced by physiological behaviors in children such as direct finger or hand sucking and higher exposure per unit of body weight [15,48]. It has been reported that the mean IQ (Intelligence Quotient) of children aged between 3 and 4 years with significantly higher blood Pb levels in the exposure area was significantly lower than that of children of the same age in the control area [31,118]. 

The carcinogenic risks (*CRs*) for As, Cr and Ni were assessed (Table 4), and Cd, Hg, Pb, Cu and Zn were excluded owing to the lack of carcinogenic slope factors. The values of *CR*_ing_, *CR*_inh_ and *CR*_derm_ ranked in the descended order of *CR*_ing_ > *CR*_dermal_ > *CR*_inh_, indicating that ingestion was the main exposure pathway that posed carcinogenic risks to human health, which was consistent with the non-carcinogenic risks. The values of *CR*_ing_ for children exceeded those of that for adults, while for *CR*_inh_ and *CR*_derm_, the values for adults were higher than for children. The mean *CR* values of As, Cr and Ni for children and adults all lay within the range of 10^−6^ to 10^−4^, indicating an acceptable carcinogenic risk to human beings. Nevertheless, due to the cumulativity of the heavy metals, the carcinogenic risks of those metals cannot be neglected, especially for children. The *CR* values for children surpassed those for the adults, which were similar to the *HI* values. The *CR* values of the three metals had a trend of Ni > Cr > As for both adults and children, which was consistent with previous research in 32 urban soils in China conducted by Pan [31].

In terms of the single urban site, the carcinogenic risks of the three assessed metals for adults were all at negligible or acceptable risk levels; however, for children, the risk values in several sites surpassed 1 × 10^−4^. The distribution of *CR* for children in urban soils in China were mapped (Figure 6). According to Figure 6, all of the *CRs* of As for children were at acceptable risk levels; however, some values were close to the threshold value of 1 × 10^−4^, illustrating that the carcinogenic risk of As should also be brought to forefront. With regard to Cr and Ni, the majority of the urban sites were at acceptable risk levels for children, with the exception of several sites that reached high risk levels for developing cancer. For Cr, the high risk sites for developing cancer for children were located in Panzhihua (1.32 × 10^−4^), Xuzhou (1.24 × 10^−4^), Xining (1.03 × 10^−4^) and Jiaozuo (2.14 × 10^−4^); for Ni, such sites were distributed in Jinchang (6.86 × 10^−4^) and Panzhihua (2.38 × 10^−4^); all of the *CR* values of the aforementioned sites exceeded the threshold value of 1 × 10^−4^, suggesting that the pollution sources should be quantitively identified and that precautionary measures should be taken in these cities to reduce the cancer risks. Additionally, children’s living behaviors should be−also properly guided, such as by encouraging them to wash their hands regularly and reducing the frequency of outdoor activities. 

## 4. Conclusions

The mean values of eight heavy metals all exceeded the soil background values in China; in particular, Cd and Hg were 8.14 and 4.15 times higher than the national background. All of the CV values of metals showed a high degree of variability and the strong influence of human activities. The hot-spot cities for heavy metals in urban soils were mainly concentrated in the southwest, southcentral, southeast coast, northcentral and northwest regions of China, based on the ordinary kriging interpolation method. The mean pollution levels of Hg and Cd represented moderately contamination levels, Pb, Cu and Zn indicated uncontaminated to moderately contaminated levels, and the levels of Cr, Ni and As represented non−contamination levels, based on the *Igeo* classification criterion. According to the *PI* classification criterion, Cd and Hg reached high contamination levels, and the other metals reached moderate contamination levels. The integral urban soils in the study areas had a high contamination level and moderate ecological risk degree, respectively. Cd and Hg posed as the dominant ecological risk factors among the eight metals, which should be labeled as priority metals for control in the urban soils around China. The high contamination and ecological risk regions were mainly concentrated in the southwest, south center, north center, east and east coast regions of China, while the hot−spot cities were mainly located in Kashi, Changsha, Shangluo, Tongchuan, Lvliang, Jinchang, Luoyang, Baoji, Ganzhou, Shenyang and Baiyin. According to the HQ values, ingestion was the dominant exposure pathway for having adverse effects on human health. Although the mean *HI* values of the eight heavy metals all showed that adverse effects on human health were unlikely and the mean *CR* values of As, Cr and Ni for children and adults all suggested an acceptable carcinogenic risk to human beings, the risks in some of the single sites should be paid significant attention. Children are exposed to more serious non-carcinogenic and carcinogenic health threats compared to adults, indicating that children’s behaviors should be adapted by reasonable guidance and paid more attention in comparison with those of adults.

### Limitations and Future Work

Due to the limited online data regarding heavy metals in urban soil in the western areas of China and the lack of data, especially for the small cities in some provinces, this study may not fully reflect the overall situation of heavy metal pollution in urban soil in China. With the action plan on The Belt and Road proposed by the Chinese government and the number of tourists into the western part increasing year on year, heavy metal pollution of the urban soil will inevitably occur, so comprehensive further research into the urban soil of the western part is needed. 

The bioavailability is related to the morphology of the heavy metal, so the human risk assessment may be overestimated by only taking into account the total concentrations. For lack in the parameters of Cd, Hg, Pb, Cu and Zn, the hazard of which were not assessed. In addition, the elderly and sensitive humans should be studied separately, according to their body situations. Moreover, it has been reported that synergistic interactions generated by mixed metals can result in numerous adverse health effects on humans; thus, the health risks posed by multiple heavy metals should be furtherly evaluated.

Finally, at present, many soil remediation projects have been carried out in mining areas and agricultural soils; however, there is still a lack of effective treatment in urban soils around China. Therefore, soil assessment and remediation in densely populated urban areas should be extensively emphasized in the future. Additionally, the temporal and spatial variation and quantitative source apportionment of the urban soil pollution should be further studied based on the models.

## Figures and Tables

**Figure 1 ijerph-17-03099-f001:**
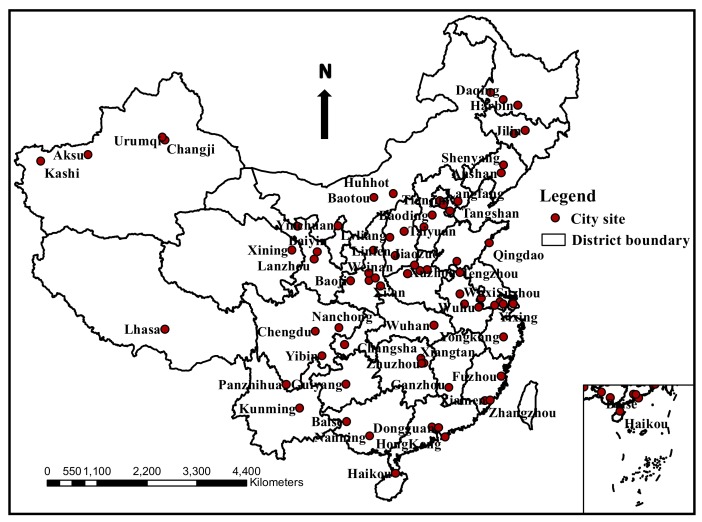
The distribution of city sites reviewed in this study around China.

**Figure 2 ijerph-17-03099-f002:**
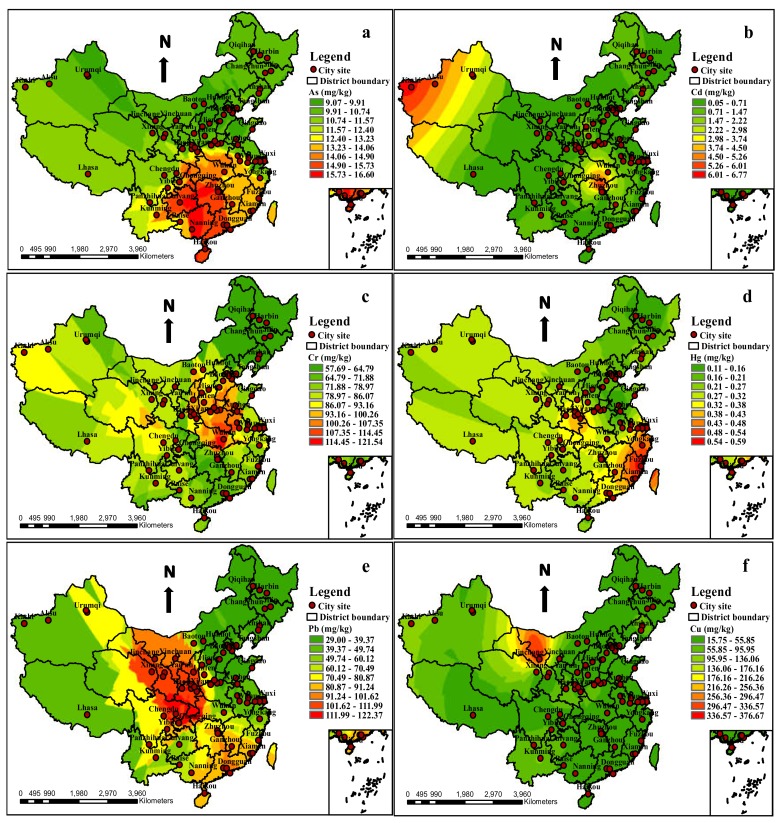
The spatial distribution of heavy metal contents in urban soils around China. (**a**): The spatial distribution of As contents in urban soils around China. (**b**): The spatial distribution of Cd contents in urban soils around China. (**c**): The spatial distribution of Cr contents in urban soils around China. (**d)**: The spatial distribution of Hg contents in urban soils around China. (**e**): The spatial distribution of Pb contents in urban soils around China. (**f**): The spatial distribution of Cu contents in urban soils around China. (**g**): The spatial distribution of Zn contents in urban soils around China. (**h**): The spatial distribution of Ni contents in urban soils around China.

**Figure 3 ijerph-17-03099-f003:**
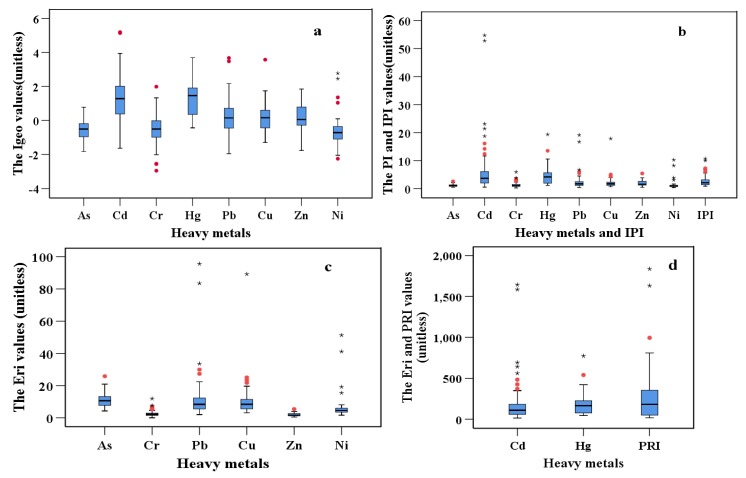
Box-plots of the geoaccumulation index (*Igeo*), pollution index (*PI*), integrated pollution index (*IPI*), ecological risk ($E_{r}^{i}$) and potential ecological risk index (*PRI*) of heavy metals in urban soil around China. The top and bottom edges of the boxes represent the 75th and 25th percentiles, respectively; the upper and lower limits of the whiskers represent the maxima and minima, excluding the outliers; the black horizontal lines inside each box represent the median values; the red dots represent mild outliers; and the black stars represent extreme outliers. (**a**): Box-plots of *Igeo* of heavy metals in urban soil around China. (**b**): Box-plots of *PI* and *IPI* of heavy metals in urban soil around China. (**c**): Box-plots of $E_{r}^{i}$ of heavy metals in urban soil around China. (**d**): Box-plots of $E_{r}^{i}$ and *PRI* of heavy metals in urban soil around China.

**Figure 4 ijerph-17-03099-f004:**
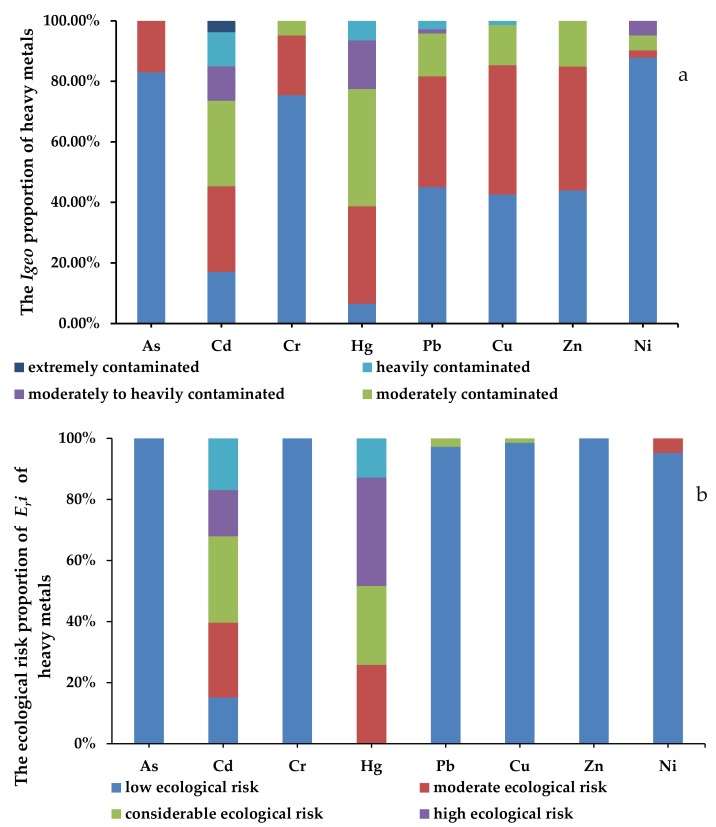
The proportion of the geoaccumulation index (Igeo) and ecological risk ($E_{r}^{i}$) of heavy metals in urban soils around China. (**a**): The proportion of Igeo heavy metals in urban soils around China. (**b**): The proportion of $E_{r}^{i}$ heavy metals in urban soils around China.

**Figure 5 ijerph-17-03099-f005:**
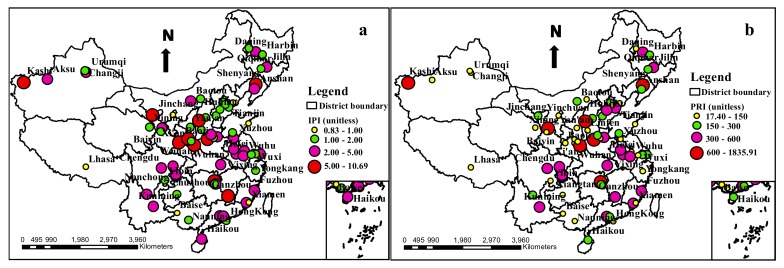
The distribution of integrated pollution index (*IPI*) and potential ecological risk index (*PRI*) of heavy metals in urban soils around China. (**a**): The distribution of *IPI* of heavy metals in urban soils around China. (**b**): The distribution of *PRI* of heavy metals in urban soils around China.

**Figure 6 ijerph-17-03099-f006:**
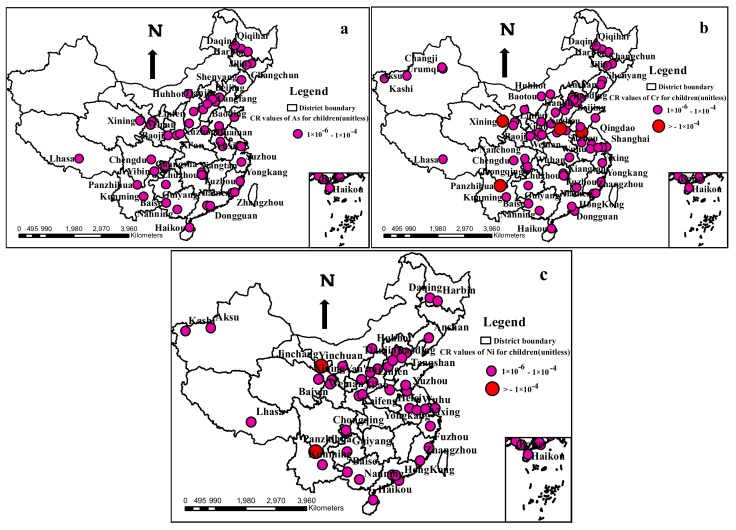
The distribution of the carcinogenic risks (*CR*) of As, Cr and Ni for children in urban soils around China. (**a**): The distribution of *CR* of As for children in urban soils around China. (**b**): The distribution of *CR* of Cr for children in urban soils around China. (**c**): The distribution of *CR* of Ni for children in urban soils around China.

**Table 1 ijerph-17-03099-t001:** Parameters for the exposure risk calculations for heavy metals.

Parameter	Implication		Children	Adult	Unit
C	concentration of the contaminant				mg kg^−1^
*IngR* ^a^	ingestion rate of soil		200	100	mg d^−1^
*InhR* ^b^	inhalation rate of soil		5	20	m^3^ d^−1^
*CF* ^a^	conversion factor		1 × 10^−6^	1 × 10^−6^	kg mg^−1^
*EF* ^b^	exposure frequency		350	350	d a^−1^
*ED* ^a^	exposure duration		6	24	y
*BW* ^b^	average body weight		15	55.9	kg
*AT* ^a^	average time	for non-carcinogens	365 × ED	365 × ED	d
		for carcinogens	365 × 70	365 × 70	d
*SA* ^b^	exposure skin surface area		1600	4350	cm^2^
*SL* ^b^	skin adherence factor		1	1	mg cm^−2^
*ABS* ^b^	dermal absorption factor		0.001	0.001	unitless
*PEF* ^b^	particle emission factor		1.32 × 10^9^	1.32 × 10^9^	m^3^ kg^−1^

^a^ USEPA 2001, ^b^ Environmental site assessment guideline in China.

**Table 2 ijerph-17-03099-t002:** The reference doses for noncarcinogenic metals (mg kg^−1^ d^−1^) and slope factors for carcinogenic metals (kg d mg^−1^) [15,48].

Metals	*RfD_ing_*	*RfD_inh_*	*RfD_derm_*	*SF_ing_*	*SF_inh_*	*SF_derm_*
As	3.00 × 10^−4^	3.10 × 10^−4^	1.20 × 10^−4^	1.50	1.51	3.66
Cd	1.00 × 10^−3^	1.00 × 10^−3^	1.00 × 10^−5^			
Cr	3.00 × 10^−3^	2.86 × 10^−5^	6.00 × 10^−5^	0.50	4.20	2.00
Hg	3.00 × 10^−4^	8.57 × 10^−5^	2.10 × 10^−5^			
Pb	3.50 × 10^−3^	3.52 × 10^−3^	5.30 × 10^−4^			
Cu	4.00 × 10^−2^	4.02 × 10^−2^	1.20 × 10^−2^			
Zn	3.00 × 10^−1^	3.00 × 10^−1^	6.00 × 10^−2^			
Ni	2.00 × 10^−2^	2.06 × 10^−2^	5.40 × 10^−4^	1.70	0.90	4.25

**Table 3 ijerph-17-03099-t003:** The concentrations of heavy metals in urban soils around China (mg/kg).

City	n	As	Cd	Cr	Hg	Pb	Cu	Zn	Ni	Reference
Beijing	550	8.55	0.17	60.30	0.32	33.70	31.30	83.80	23.30	[50]
Kaifeng	99	6.31	1.05	53.11	—	36.71	36.40	164.03	23.87	[51]
Guiyang	62	20.53	0.32	35.71	0.19	22.17	64.87	217.90	48.65	[52]
Shijiazhuang	220	9.42	0.28	71.85	0.11	31.00	27.39	104.48	28.20	[53]
Chongqing	48	8.02	0.98	26.58	0.31	32.61	24.63	96.77	25.64	[54]
Luoyang	215	—	1.71	71.42	—	65.92	85.40	215.75	—	[55]
Jilin	136	11.32	0.20	80.40	0.19	34.70	24.70	109.20	—	[56]
Tangshan	63	6.79	0.10	46.20	0.07	25.08	20.97	63.38	17.33	[57]
Guangzhou	426	17.40	0.32	—	0.61	87.60	35.80	107.00	18.70	[58]
Changchun	352	12.50	0.13	66.00	0.12	35.40	29.40	90.00	—	[59]
Huhhot	62	6.40	—	54.75	—	11.63	30.07	89.93	16.47	[60]
Xi’an	62	12.20	—	69.80	—	36.90	32.40	101.30	30.70	[61]
Yongkang	181	6.00	—	121.00	—	40.00	24.00	95.00	23.00	[62]
Shanghai	273	—	0.52	107.90	—	70.69	59.25	301.40	31.14	[63]
Zhengzhou	90	—	—	—	—	39.63	59.11	91.67	—	[64]
Wuhan	467	—	3.22	41.85	—	28.16	18.82	88.07	—	[65]
Chengdu	35	11.00	0.36	60.00	0.31	76.90	42.00	224.00	—	[66]
Shenyang	93	22.69	1.10	67.90	0.39	116.76	92.45	234.80	—	[24]
HongKong	152	—	0.62	23.10	—	94.60	23.30	125.00	12.40	[29]
Baotou	88	—	0.29	35.60	—	39.82	27.76	79.45	—	[67]
Qingdao	83	7.71	0.11	55.83	—	27.11	17.95	49.66	—	[68]
Baiyin	132	5.91	0.29	52.62	0.27	64.59	57.33	197.06	17.30	[69]
Taiyuan	80	10.96	0.21	73.69	0.12	26.29	28.87	86.08	29.76	[70]
Changsha	110	32.80	6.90	121.00	0.41	89.40	51.40	276.00	—	[71]
Baoji	50	8.75	—	98.08	—	409.20	107.19	374.47	—	[72]
Urumqi	85	—	—	69.24	—	28.74	47.90	263.24	—	[73]
Xuzhou	172	17.10	0.40	219.40	0.05	35.50	43.80	163.80	42.90	[74]
Nanjing	150	—	—	84.70	—	107.30	66.10	162.60	—	[75]
Wuxi	1957	9.77	0.17	—	0.38	41.76	34.73	86.95	—	[76]
Zhuzhou	60	20.68	0.61	91.00	0.21	86.00	37.40	141.00	—	[77]
Daqing	—	5.70	1.00	98.10	—	26.20	—	39.93	42.08	[78]
Suzhou	167	15.51	0.33	75.60	0.52	40.26	—	—	—	[79]
Hefei	151	10.80	0.20	—	0.18	37.00	38.60	108.80	27.30	[80]
Xiangtan	54	18.72	0.46	84.00	0.24	65.00	37.50	127.00	—	[81]
Zhangzhou	108	6.86	0.35	29.70	0.47	75.90	32.60	106.70	12.80	[82]
Kunming	204	13.23	1.32	109.94	0.24	60.28	111.25	150.63	50.13	[83]
Lanzhou	117	20.63	—	93.96	—	42.72	52.41	184.22	38.22	[84]
Xining	155	7.65	—	182.54	—	32.57	19.73	45.13	25.21	[85]
Lhasa	—	9.66	0.26	19.69	—	16.22	13.55	102.84	14.83	[86]
Haikou	70	3.82	0.25	92.40	0.07	29.10	26.70	84.10	52.50	[87]
Panzhihua	17	9.67	0.30	234.04	—	30.01	62.86	158.22	125.10	[88]
Qiqihar	55	7.87	—	34.85	—	13.37	—	62.08	—	[89]
Jinchang	74	—	0.30	—	—	32.20	430.00	116.00	361.00	[90]
Harbin	307	8.87	0.17	61.28	0.08	26.74	22.33	72.03	25.73	[91]
Yinchuan	96	—	—	109.10	—	25.00	16.80	26.00	25.30	[92]
Wuhu	153	—	1.20	78.30	—	29.10	35.00	96.80	26.30	[93]
Jiaozuo	44	—	0.33	378.86	—	20.23	36.26	—	—	[94]
Anshan	115	—	0.86	69.90	—	45.10	52.30	213.00	33.50	[19]
Tianjin	70	11.00	0.18	51.00	0.43	45.00	33.00	148.00	39.00	[28]
Huainan	36	12.54	0.19	49.39	0.21	24.21	21.74	—	—	[15]
Yan’an	40	—	0.10	66.22	—	20.18	23.65	71.20	37.56	[95]
Fuzhou	179	8.28	0.30	40.11	0.77	89.83	39.41	158.65	16.04	[96]
Linfen	217	16.59	—	41.12	—	33.91	16.60	101.32	36.11	[97]
Baise	36	8.68	0.16	67.19	—	20.74	29.04	86.01	24.04	[98]
Dongguan	170	13.33	0.25	74.90	0.15	160.27	66.64	150.80	44.45	[99]
Yixing	47	—	0.11	65.23	—	31.39	23.57	68.62	26.40	[100]
Xiamen	146	5.82	—	41.77	—	37.35	23.26	—	—	[14]
Kashi	—	—	6.34	29.33	—	81.39	103.54	140.7	15.95	[101]
Aksu	50	—	—	25.56	—	75.98	97.26	120.33	8.41	[102]
Yibin	47	8.70	—	—	—	65.20	61.50	135.40	—	[103]
Nanchong	—	—	0.82	15.32	—	85.99	50.29	106.83	—	[104]
Shangluo	15	—	0.55	—	0.58	127.99	59.31	155.08	—	[105]
Tongchuan	26	—	—	93.48	—	357.47	66.48	116.17	—	[106]
Lvliang	—	—	2.74	—	—	70.70	18.71	47.95	18.95	[107]
Tengzhou	335	7.83	0.14	60.37	0.05	25.10	27.02	66.09	26.95	[108]
Ganzhou	50	—	1.74	173.17	—	216.99	32.03	146.61	—	[109]
Langfang	—	7.58	0.16	63.30	0.04	27.10	20.10	72.20	—	[110]
Baoding	48	9.75	0.18	66.30	0.21	38.99	29.10	122.36	28.05	[111]
Weinan	38	8.49	—	96.99	—	46.71	20.88	71.56	25.43	[112]
Changji	35	—	—	52.18	—	7.52	65.99	221.48	—	[113]
Nanning	46	23.20	0.77	20.94	—	65.56	45.56	105.41	37.04	[114]
Min		3.82	0.10	15.32	0.04	7.52	13.55	26.00	8.41	
Max		32.80	6.90	378.86	0.77	409.20	430.00	374.47	361.00	
Median		9.67	0.32	66.30	0.21	37.35	34.87	107.00	26.40	
Mean±SD		11.53±5.79	0.79±1.32	77.86±57.67	0.27±0.19	60.26±66.02	47.72±52.55	128.21±66.49	37.99±53.61	
CV(%)		50	168	75	69	111	112	52	144	
BGV of China		11.2	0.097	61	0.065	26	22.6	74.2	26.9	[35]

**Table 4 ijerph-17-03099-t004:** The mean values of the human health risks for children and adults posed by each element, according to exposure pathway.

		As	Cd	Cr	Hg	Pb	Cu	Zn	Ni
HQ_ing_	children	4.98 × 10^−1^	1.03 × 10^−2^	3.36 × 10^−1^	1.16 × 10^−2^	2.23 × 10^−1^	1.55 × 10^−2^	5.54 × 10^−3^	2.46 × 10^−2^
	adults	6.69 × 10^−2^	1.38 × 10^−3^	4.51 × 10^−2^	1.55 × 10^−3^	2.99 × 10^−2^	2.08 × 10^−3^	7.43 × 10^−4^	3.30 × 10^−3^
HQ_inh_	children	9.01 × 10^−6^	1.92 × 10^−7^	6.59 × 10^−4^	7.56 × 10^−7^	4.15 × 10^−6^	2.87 × 10^−7^	1.03 × 10^−7^	4.47 × 10^−7^
	adults	9.67 × 10^−6^	2.07 × 10^−7^	7.08 × 10^−4^	8.12 × 10^−7^	4.45 × 10^−6^	3.09 × 10^−7^	1.11 × 10^−7^	4.79 × 10^−7^
HQ_derm_	children	9.83 × 10^−3^	8.13 × 10^−3^	1.33 × 10^−1^	1.30 × 10^−3^	1.16 × 10^−2^	4.07 × 10^−4^	2.19 × 10^−4^	7.20 × 10^−3^
	adults	7.17 × 10^−3^	5.93 × 10^−3^	9.68 × 10^−2^	9.51 × 10^−4^	8.48 × 10^−3^	2.97 × 10^−4^	1.59 × 10^−4^	5.25 × 10^−3^
*HI*	children	5.08 × 10^−1^	1.84 × 10^−2^	4.70 × 10^−1^	1.29 × 10^−2^	2.35 × 10^−1^	1.59 × 10^−2^	5.76 × 10^−3^	3.18 × 10^−2^
	adults	7.40 × 10^−2^	7.31 × 10^−3^	1.43 × 10^−1^	2.50 × 10^−3^	3.84 × 10^−2^	2.37 × 10^−3^	9.03 × 10^−4^	8.56 × 10^−3^
CR_ing_	children	1.90 × 10^−5^		4.27 × 10^−5^					7.08 × 10^−5^
	adults	1.02 × 10^−5^		2.29 × 10^−5^					3.80 × 10^−5^
CR_inh_	children	3.61 × 10^−10^		6.79 × 10^−9^					7.10 × 10^−10^
	adults	1.55 × 10^−9^		2.91 × 10^−8^					3.05 × 10^−9^
CR_derm_	children	3.70 × 10^−7^		1.37 × 10^−6^					1.42 × 10^−6^
	adults	1.08 × 10^−6^		3.98 × 10^−6^					4.13 × 10^−6^
*CR*	children	1.93 × 10^−5^		4.40 × 10^−5^					7.22 × 10^−5^
	adults	1.13 × 10^−5^		2.69 × 10^−5^					4.21 × 10^−5^

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
