# Peer review of "Concentration, Spatial Distribution, Contamination Degree and Human Health Risk Assessment of Heavy Metals in Urban Soils across China between 2003 and 2019—A Systematic Review"

_ijerph, 2020, doi:10.3390/ijerph17093099_

Round 1
Reviewer 1 Report
The manuscript "Concentration, spatial distribution, contamination degree and human health risk assessment of heavy metals in urban soil across China between 2003 and 2019—A Systematic Review" by Shuangmei Tong et al provided the information about heavy metals pollution such as As, Cd, Cr, Hg, Pb, Cu, Zn and Ni in the 76 urban soils throughout China based on the online literature data. The topic of the study fits the scope of the International Journal of Environmental Research and Public Health and the manuscript will be interesting for experts in this research area as it provides systematic overview data on the assessment the accumulation and spatial distributions of heavy metals in urban soil across China.
However, some revisions should be performed before publication.
Table 1 is not necessary to place in the text of the article. The authors need provide clear data selection criteria in the text of the section "2.1.1. The Searching Method".
The authors have used data from various sources (about 5750 articles). How was unified methods for determining the content of HM? It is necessary to point what used method of heavy metals determining concentrations.
Please check in the appropriate sections what was taken as the background value for the calculation of the Pollution index, Potential ecological risk and health risk assessment? Authors should be mention what parameters were calculated in section 2.2.4. Statistical analysis.
The human risks assessment may underestimated only taking into account the total concentrations of heavy metals. Present study is based only at total metal concentrations. However, risks to human health and the environment must be calculated based on the mobile forms of metals. They determines the ecological consequences of the metal pollution of soils, because certain solid mineral and organic soil phases have a high buffering capacity for inorganic contaminants, thus providing a protective function for soil. Authors must provide the data about mobile forms of metals.
Reviewer 2 Report
The article entitled “Concentration, spatial distribution, contamination degree and human health risk assessment of heavy metals in urban soil across China between 2003 and 2019—A Systematic Review” by Shuangmei Tong , Hairong Li * , Wang Li , Muyessar Turdi , Linsheng Yang has been reviewed.
Detailed comments are included in the attached document.
I am not an expert in English, but it seems to me that the article should be checked for linguistic correctness.
Although the topic taken in the article is local (concerns soil pollution with heavy metals in China), the article may be of interest to a wider audience. Therefore, I think that after corrections, the article can be considered for publication in IJERPH.

Author Response
Dear reviewer,
Thank for your comments.
The grammar and language have been checked and corrected, and the comments in the attached document has been corrected, the detailed modifications are included in the revision. But one problem that I don't know how to modify the key words, because the words in the title are also the main researsh of the artical.
Best regards.
Reviewer 3 Report
The work is good.
Please pay attention to the paper presentation/formatting. It needs corrections. It looks sloppy.
It is difficult to follow the text, tables and figures. There are too many numbers in the text, poorly separated. Keep tables, eliminate the numbers and just explain the results. The tables and figures are thrown somewhere in the text.
Correct repetitive English.
Author Response
The paper presentation/formatting has been checked and corrected.
The tables and figures have been transferred to the appropriate place, and some numbers have been deleted.
The detailed modifications are seen in the rivision.
Round 2
Reviewer 1 Report
This manuscript was considerably improved. Therefore, I recommend acceptation of this manuscript for publication in the International Journal of Environmental Research and Public Health.
Author Response
Thanks for your comment.
Best regards.
Reviewer 3 Report
Dear Authors,
It is a good job. Thus, it is a pity for not doing it more accessible for the readers.
Please see the document and my comments in the text.
Best regards

Author Response
Thanks for your comments, I have revised my paper, please see the details in the text.
Best regards